# VC DIMENSION OF PARTIALLY QUANTIZED NEURAL NETWORKS IN THE OVERPARAMETRIZED REGIME

**Yutong Wang[1] & Clayton Scott[1,2]**
[1]Department of Electrical Engineering and Computer Science
[2]Department of Statistics
University of Michigan
Ann Arbor, MI 48109, USA
`{yutongw,clayscot}@umich.edu`

## ABSTRACT

Vapnik-Chervonenkis (VC) theory has so far been unable to explain the small generalization error of overparametrized neural networks. Indeed, existing applications of VC theory to large networks obtain upper bounds on VC dimension that are proportional to the number of weights, and for a large class of networks, these upper bound are known to be tight. In this work, we focus on a subclass of partially quantized networks that we refer to as *hyperplane arrangement neural networks* (HANNs). Using a sample compression analysis, we show that HANNs can have VC dimension significantly smaller than the number of weights, while being highly expressive. In particular, empirical risk minimization over HANNs in the overparametrized regime achieves the minimax rate for classification with Lipschitz posterior class probability. We further demonstrate the expressivity of HANNs empirically. On a panel of 121 UCI datasets, overparametrized HANNs match the performance of state-of-the-art full-precision models.

## 1 INTRODUCTION

Neural networks have become an indispensable tool for machine learning practitioners, owing to their impressive performance especially in vision and natural language processing (Goodfellow et al., 2016). In practice, neural networks are often applied in the *overparametrized* regime and are capable of fitting even random labels (Zhang et al., 2021). Evidently, these overparametrized models perform well on real world data despite their ability to grossly overfit, a phenomenon that has been dubbed "the generalization puzzle" (Nagarajan & Kolter, 2019).

Toward solving this puzzle, several research directions have flourished and offer potential explanations, including implicit regularization (Chizat & Bach, 2020), interpolation (Chatterji & Long, 2021), and benign overfitting (Bartlett et al., 2020). So far, VC theory has not been able to explain the puzzle, because existing bounds on the VC dimensions of neural networks are on the order of the number of weights (Maass, 1994; Bartlett et al., 2019). It remains unknown whether there exist neural network architectures capable of modeling rich set of classfiers with low VC dimension.

The focus of this work is on a class of neural networks with threshold activation that we refer to as *hyperplane arrangement neural networks* (HANNs). Using the theory of sample compression schemes (Littlestone & Warmuth, 1986), we show that HANNs can have VC dimension that is significantly smaller than the number of parameters. Furthermore, we apply this result to show that HANNs have high expressivity by proving that HANN classifiers achieve minimax-optimality when the data has Lipschitz posterior class probability in an overparametrized setting.

We benchmark the empirical performance of HANNs on a panel of 121 UCI datasets, following several recent neural network and neural tangent kernel works (Klambauer et al., 2017; Wu et al., 2018; Arora et al., 2019; Shankar et al., 2020). In particular, Klambauer et al. (2017) showed that, using a properly chosen activation, overparametrized neural networks perform competitively compared to classical shallow methods on this panel of datasets. Our experiments show that HANNs, a partially-quantized model, match the classification accuracy of the self-normalizing neural network

(Klambauer et al., 2017) and the dendritic neural network (Wu et al., 2018), both of which are full-precision models.

## 1.1 RELATED WORK

**VC dimensions of neural networks.** The *VC dimension* (Vapnik & Chervonenkis, 1971) is a combinatorial measure of the complexity of a concept class, i.e., a set of classifiers. The *Fundamental Theorem of Statistical Learning* (Shalev-Shwartz & Ben-David, 2014, Theorem 6.8) states that a concept class has finite VC-dimension if and only if it is probably approximately correct (PAC) learnable, where the VC-dimension is tightly related to the number of samples required for PAC learning.

For threshold networks, Cover (1965); Baum & Haussler (1989) showed a VC-dimension upper bounded of $O(w \log w)$, where $w$ is the number of parameters. Maass (1994) obtained a matching *lower* bound attained by a network architecture with two hidden layers. More recently, Bartlett et al. (2019) obtained the upper and lower bounds $O(w\ell \log w)$ and $\Omega(w\ell \log(w/\ell))$ respectively for the case when the activation is piecewise linear, where $\ell$ is the number of layers. These lower bounds are achieved by somewhat unconventional network architectures. The architectures we consider exclude these and thus we are able to achieve a smaller upper bound on the VC dimensions.

**The generalization puzzle.** In practice, neural networks that achieve state-of-the-art performance use significantly more parameters than samples, a phenomenon that cannot be explained by classical VC theory if the VC dimension $\geq$ number of weights. This has been dubbed the *generalization puzzle* (Zhang et al., 2021). To explain the puzzle, researchers have pursued new directions including margin-based bounds (Neyshabur et al., 2017; Bartlett et al., 2017), PAC-Bayes bounds (Dziugaite & Roy, 2017), and implicit bias of optimization methods (Gunasekar et al., 2018; Chizat & Bach, 2020). We refer the reader to the recent article by Bartlett et al. (2021) for a comprehensive coverage of this growing literature.

The generalization puzzle is *not* specific to deep learning. For instance, AdaBoost has been observed to continue to decrease the test error while the VC dimension grows linearly with the number of boosting rounds (Schapire, 2013). Other learning algorithms that exhibits similarly surprising behavior include random forests (Wyner et al., 2017) and kernel methods (Belkin et al., 2018).

**Minimax-optimality.** Whereas VC theory is distribution-independent, minimax theory is concerned with the question of *optimal* estimation/classification under distributional assumptions[1].

A minimax optimality result shows that the expected excess classification error goes to zero at the fastest rate possible, as the sample size tend to infinity. For neural networks, this often involves a hyperparameter selection scheme in terms of the sample size.

Faragó & Lugosi (1993) show minimax-optimality of (underparametrized) neural networks for learning to classify under certain assumptions on the Fourier transform of the data distribution. Schmidt-Hieber (2020) shows minimax-optimality of *s-sparse* neural networks for regression over Hölder classes, where at most $s = O(n \log n)$ network weights are nonzero, and $n$ = the number of training samples. Kim et al. (2021) extends the results of Schmidt-Hieber (2020) to the classification setting, remarking that effective optimization under sparsity constraint is lacking. Kohler & Langer (2020) and Langer (2021) proved minimax-optimality without the sparsity assumption, however in an underparametrized setting. To the best of our knowledge, our result is the first to establish minimax optimality of overparametrized neural networks without a sparsity assumption.

**(Partially) quantized neural networks.** Quantizing some of the weights and/or activations of neural networks has the potential to reduce the high computational burden of neural networks at test time (Qin et al., 2020). Many works have focused on the efficient training of quantized neural networks to close the performance gap with full-precision architectures (Hubara et al., 2017; Rastegari et al., 2016; Lin et al., 2017). Several works have observed that quantization of the activations, rather than of the weights, leads to a larger accuracy gap (Cai et al., 2017; Mishra et al., 2018; Kim et al., 2019).

Towards explaining this phenomenon, researchers have focused on understanding the so-called *coarse gradient*, a term coined by Yin et al. (2019), often used in training QNNs as a surrogate

---

[1]The No-Free-Lunch Theorem (Devroye, 1982) implies that no classifier can be minimax optimal without distributional assumption.

for the usual gradient. One commonly used heuristic is the *straight-through-estimator* (STE) first introduced in an online course by Hinton et al. (2012). Theory supporting the STE heuristic has recently been studied in Li et al. (2017) and Yin et al. (2019).

QNNs have also been analyzed from other theoretical angles, including mean-field theory (Blumenfeld et al., 2019), memory capacity (Vershynin, 2020), Boolean function representation capacity (Baldi & Vershynin, 2019) and adversarial robustness (Lin et al., 2018). Of particular relevance to our work, Maass (1994) constructed an example of a QNN architecture with VC dimension on the order of the number of weights in the network. In contrast, our work shows that there exist QNN architectures with much smaller VC dimensions.

**Sample compression schemes.** Many concept classes with geometrically structured decision regions, such as axis-parallel rectangles, can be trained on a properly chosen size $\sigma$ subset of an arbitrarily large training dataset without affecting the result. Such a concept class is said to admit a *sample compression schemes* of size $\sigma$, a notion introduced by Littlestone & Warmuth (1986) who showed that the VC dimension of the class is upper bounded by $O(\sigma)$. Furthermore, the authors posed the *Sample Compression Conjecture*. See Moran & Yehudayoff (2016) for the best known partial result and an extensive review of research in this area. Besides the conjecture, sample compression schemes have also been applied to other long-standing problems in learning theory (Hanneke et al., 2019; Bousquet et al., 2020; Ashtiani et al., 2020). To the best of our knowledge, our work is the first to apply sample compression schemes to neural networks.

## 2 NOTATIONS

The set of real numbers is denoted $\mathbb{R}$. The unit interval is denoted $[0, 1]$. For an integer $k \geq 1$, let $[k] = \{1, \ldots, k\}$. We use $\mathcal{X}$ to denote the feature space, which in this work will either be $\mathbb{R}^d$ or $[0, 1]^d$ where $d \geq 1$ is the ambient dimension/number of features.

Denote by $\mathbb{I}\{\texttt{input}\}$ the *indicator* function which returns 1 if `input` is true and 0 otherwise. The *sign* function is given by $\sigma_{\texttt{sgn}}(t) = \mathbb{I}\{t \geq 0\} - \mathbb{I}\{t < 0\}$. For vector inputs, $\sigma_{\texttt{sgn}}$ applies entry-wise.

The set of labels for binary classification is denoted $\mathbb{B} := \{\pm 1\}$. Joint distributions on $\mathcal{X} \times \mathbb{B}$ are denoted by $P$, where $X, Y \sim P$ denotes a random instance-label pair distributed according to $P$. Let $f : \mathcal{X} \to \mathbb{B}$ be a binary classifier. The *risk* with respect to $P$ is denoted by $R_P(f) := P(f(X) \neq Y)$. For an integer $n \geq 1$, the *empirical risk* is the random variable $\hat{R}_{P,n}(f) := \frac{1}{n} \sum_{i=1}^n \mathbb{I}\{f(X_i) \neq Y_i\}$, where $(X_1, Y_1), \ldots, (X_n, Y_n) \sim P$ are i.i.d. The *Bayes risk* $\inf_{f:\mathcal{X}\to\mathbb{B}} R_P(f)$ with respect to $P$ is denoted by $R_P^*$.

Let $f, g : \{1, 2, \ldots\} \to \mathbb{R}_{\geq 0}$ be nonnegative functions on the natural numbers. We write $f \asymp g$ if there exists $\alpha, \beta > 0$ such that for all $n = 1, 2, \ldots$ we have $\alpha g(n) \leq f(n) \leq \beta g(n)$.

## 3 HYPERPLANE ARRANGEMENT NEURAL NETWORKS

A hyperplane $H$ in $\mathbb{R}^d$ is specified by its normal vector $w \in \mathbb{R}^d$ and bias $b \in \mathbb{R}$. The mapping $x \mapsto \sigma_{\texttt{sgn}}(w^\top x + b)$ indicates the side of $H$ that $x$ lies on, and hence induces a partition of $\mathbb{R}^d$ into two halfspaces. A set of $k \geq 1$ hyperplanes is referred to as a *k-hyperplane arrangement*, and specified by a matrix of normal vectors and a vector of offsets:

$$\mathbf{W} = [w_1 \cdots w_k] \in \mathbb{R}^{d \times k} \quad \text{and} \quad b = [b_1, \ldots, b_k]^\top.$$

Let $q_{\mathbf{W},b}(x) := \sigma_{\texttt{sgn}}(\mathbf{W}^\top x + b)$ for all $x \in \mathbb{R}^d$. The vector $q_{\mathbf{W},b}(x) \in \mathbb{B}^k$ is called a *sign vector* and the set of all realizable sign vectors is denoted $\mathfrak{S}_{\mathbf{W},b} := \{q_{\mathbf{W},b}(x) : x \in \mathbb{R}^d\}$. Each sign vector $s \in \mathfrak{S}_{\mathbf{W},b}$ uniquely defines a set $\{x \in \mathbb{R}^d : q_{\mathbf{W},b}(x) = s\}$ known as a *cell* of the hyperplane arrangement. The set of all cells forms a partition of $\mathbb{R}^d$. For an example, see Figure 1-left.

A classical result in the theory of hyperplane arrangement due to Buck (1943) gives the following tight upper bound on the number of distinct sign patterns/cells:

$$|\mathfrak{S}_{\mathbf{W},b}| \leq \binom{k}{\leq d} := \begin{cases} 2^k & : k < d, \\ \binom{k}{0} + \binom{k}{1} + \cdots + \binom{k}{d} & : k \geq d. \end{cases} \tag{1}$$

See Fukuda (2015) Theorem 10.1 for a simple proof. A *hyperplane arrangement classifier* assigns a binary label $y \in \mathbb{B}$ to a point $x \in \mathbb{R}^d$ solely based on the sign vector $q_{\mathbf{W},b}(x)$.

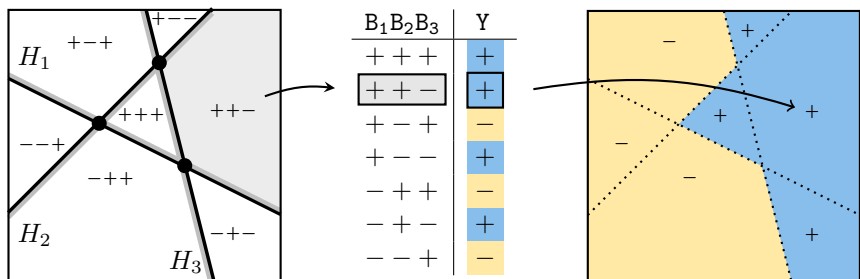

Figure 1: Left: An arrangement of 3 hyperplanes $\{H_1, H_2, H_3\}$ in $\mathbb{R}^2$. There are 7 sign patterns. Middle: An example of a lookup table (see Remark 3.2). Right: the resulting classifier.

**Definition 3.1.** Let $\mathbb{B}^{\mathcal{X}}$ be the set of all functions from $\mathcal{X}$ to $\mathbb{B}$. A *concept class* $\mathcal{C}$ over $\mathcal{X}$ is a subset of $\mathbb{B}^{\mathcal{X}}$. Fix $r, k$ positive integers, $r \leq \min\{d, k\}$. Let $\mathtt{Bool}_k$ be the set of all Boolean functions $\mathbb{B}^k \to \mathbb{B}$. The *hyperplane arrangement classifier* class is the concept class, denoted $\mathtt{HAC}(d, r, k)$, over $\mathbb{R}^d$ defined by

$$\mathtt{HAC}(d, r, k) = \{h \circ q_{\mathbf{W},b} : h \in \mathtt{Bool}_k, \; q_{\mathbf{W},b}(x) := \sigma_{\mathtt{sgn}}(\mathbf{W}^\top x + b),$$
$$\mathbf{W} \in \mathbb{R}^{d \times k}, \; \mathrm{rank}(\mathbf{W}) \leq r, \; b \in \mathbb{R}^k\}.$$

See Figure 2 for a graphical representation of $\mathtt{HAC}(d, r, k)$. When the set of Boolean functions is realized by a neural network, we refer to the resulting classifier as a *hyperplane arrangement neural network* (HANN).

*Remark* 3.2. Consider a fixed hyperplane arrangement $\mathbf{W}, b$ and Boolean function $h \in \mathtt{Bool}_k$. When performing prediction with the classifer $h \circ q_{\mathbf{W},b}$, the feature vector $x$ is mapped to a sign vector to which $h$ is applied. Thus, we do not need to know how $h$ behaves outside of $\mathfrak{S}_{\mathbf{W},b}$. The restriction of $h$ to $\mathfrak{S}_{\mathbf{W},b}$ is a *partially defined Boolean function* or a *lookup table*.

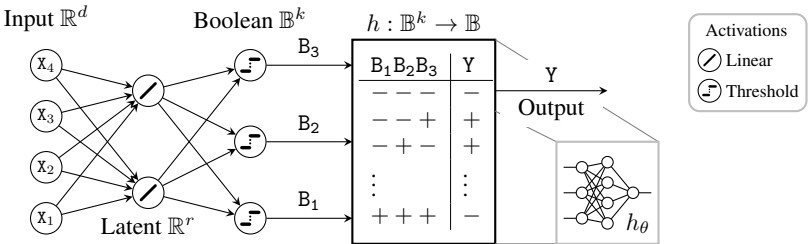

Figure 2: The $\mathtt{HAC}(d, r, k)$ concept class as a neural network where $d = 4$, $r = 2$ and $k = 3$. The Boolean function $h$ is realized as a neural network $h_\theta$.

*Remark* 3.3. The hidden layer of width $r$ in Figure 2 allows the user to impose the restriction that the hyperplane arrangement classifier depends only on $r$ relevant features, which can be either learned or defined by data preprocessing. When $r = d$, no restriction is imposed. In this case, the input layer is directly connected to the Boolean layer. This is consistent with Definition 3.1 where the rank constraint $\mathrm{rank}(\mathbf{W}) \leq r$ becomes trivial.

Our next goal is to upper bound the VC dimension of $\mathtt{HAC}(d, r, k)$.

**Definition 3.4** (VC-dimension). Let $\mathcal{C} \subseteq \mathbb{B}^{\mathcal{X}}$ be a concept class over $\mathcal{X}$. A set $S := \{x_1, \ldots, x_n\} \subseteq \mathcal{X}$ is *shattered* by $\mathcal{C}$ if for all sequences $(y_1, \ldots, y_n) \in \mathbb{B}^n$, there exists $f \in \mathcal{C}$ such that $f(x_i) = y_i$ for all $i \in [n]$. The *VC-dimension* of $\mathcal{C}$ is defined as

$$\mathtt{VC}(\mathcal{C}) = \sup\{|S| : S \subseteq \mathcal{X}, \; S \text{ is shattered by } \mathcal{C}\}.$$

The VC-dimension has many far-reaching consequences in learning theory and, in particular, classification. One of these consequences is a sufficient (in fact also necessary) condition for *uniform convergence* in the sense of the following well-known theorem. See Shalev-Shwartz & Ben-David (2014) Theorem 6.8.

**Theorem 3.5.** *Let $\mathcal{C}$ be a concept class over $\mathcal{X}$. There exists a constant $C > 0$ such that for all joint distributions $P$ on $\mathcal{X} \times \mathbb{B}$ and all $f \in \mathcal{C}$, we have $|\hat{R}_{P,n}(f) - R_P(f)| \leq C\sqrt{(\text{VC}(\mathcal{C}) + \log(1/\delta))/n}$ with probability at least $1 - \delta$ with respect to the draw of $(X_1, Y_1), \ldots, (X_n, Y_n)$.*

The above VC bound is useless in the overparametrized setting if $\text{VC}(\mathcal{C}) = \Theta(\#$ of weights$)$ because $\#$ of weights $> n$ and therefore the VC bound does not vanish. We now present our main result: an upper bound on the VC dimension of $\text{HAC}(d, r, k)$.

**Theorem 3.6.** *Let $d, r, k \geq 1$ be integers and $\text{HAC}(d, r, k)$ be defined as in Definition 3.1. Then*

$$\text{VC}(\text{HAC}(d, r, k)) \leq 8 \cdot \left( k(d+1) + k(d+1)(1 + \lceil \log_2 k \rceil) + \binom{k}{\leq r} \right).$$

In the next section, we will prove this result using a sample compression scheme. Before proceeding, we comment on the significance of the result.

*Remark* 3.7. Since $\binom{k}{\leq r} = O(k^r)$, we have $\text{VC}(\text{HAC}(d, r, k)) = O(k^r + dk \log k)$ which only involves the input dimension $d$ and the width of the first two hidden layers $r$ and $k$. For constant $d$ and $r \geq 2$, this reduces to $\text{VC}(\text{HAC}(d, r, k)) = O(k^r)$. In particular, the number of weights used by an architecture to implement the Boolean function $h$ does not affect the VC dimension at all and can be even infinitely wide such as in Neural Tangent Kernels (Jacot et al., 2018).

For instance, Mukherjee & Basu (2017) Lemma 2.1 states that a 1-hidden layer neural network with ReLU activation can model any $k$-input Boolean function if the hidden layer has width $\geq 2^k$. Note that this network uses $\geq k2^k$ weights, and $k2^k \gg k^r$ for fixed $r$ and $k$ large.

Baldi & Vershynin (2019) study implementation of Boolean functions using threshold networks. A consequence of their Theorem 9.3 is that a 2-hidden layer network with widths $\geq c2^{k/2}/\sqrt{k}$ can implement all $k$ input Boolean functions, where $c$ is a constant not depending on $k$. This requires $\geq c^2 2^k/k$ weights which again is exponentially larger than $k^r$. Furthermore, this lower bound on the weights is also necessary as $k \to \infty$.

*Remark* 3.8. Using the memorization approach of Vershynin (2020); Rajput et al. (2021), it is also possible to implement $\text{HAC}(d, r, k)$ with far fewer weights than in Remark 3.7. This can be done by applying Theorem 1 of Rajput et al. (2021) to memorize the $\binom{k}{\leq r}$ distinct sign vectors in $\mathbb{R}^k$.

## 4    A SAMPLE COMPRESSION SCHEME

In this section, we will construct a sample compression scheme for $\text{HAC}(d, r, k)$. As alluded to in the Related Work section, the size of a sample compression scheme upper bounds the VC-dimension of a concept class, which will be applied to prove Theorem 3.6. We first recall the definition of sample compression schemes with side information introduced in Littlestone & Warmuth (1986).

**Definition 4.1.** Let $\mathcal{C}$ be a concept class. A length $n$ sequence $\{(x_i, y_i) \in \mathcal{X} \times \mathbb{B}\}_{i \in [n]}$ is $\mathcal{C}$-*labelled* if there exists $f \in \mathcal{C}$ such that $f(x_i) = y_i$ for all $i \in [n]$. Denote by $L_{\mathcal{C}}(n)$ the set of $\mathcal{C}$-labelled sequences of length at most $n$. Denote by $L_{\mathcal{C}}(\infty)$ the set of all $\mathcal{C}$-labelled sequences of finite length. The concept class $\mathcal{C}$ over $\mathcal{X}$ has an $m$-*sample compression scheme with $s$-bits of side information* if there exists a pair of maps $(\rho, \kappa)$ where

$$\kappa : L_{\mathcal{C}}(\infty) \to L_{\mathcal{C}}(m) \times \mathbb{B}^s, \quad \rho : L_{\mathcal{C}}(m) \times \mathbb{B}^s \to \mathbb{B}^{\mathcal{X}}$$

such that for all $\mathcal{C}$-labelled sequences $S := \{(x_i, y_i)\}_{i \in [n]}$, we have $\rho(\kappa(S))(x_i) = y_i$ for all $i \in [n]$. The *size* of the sample compression scheme is $\text{size}(\rho, \kappa) := m + s$.

Intuitively, $\kappa$ and $\rho$ can be thought of as the *compression* and the *reconstruction* maps, respectively. The compression map $\kappa$ keeps $m$ elements from the training set and $s$ bits of additional information, which $\rho$ uses to reconstruct a classifier that correctly labels the uncompressed training set.

The main result of this section is:

**Theorem 4.2.** $\mathtt{HAC}(d, r, k)$ *has a sample compression scheme* $(\rho, \kappa)$ *of size*

$$\mathtt{size}(\rho, \kappa) = k(d+1) + k(d+1)(1 + \lceil \log_2 k \rceil) + \binom{k}{\leq r}.$$

We remark that both the hyperplane arrangement $(\mathbf{W}, b)$ and the Boolean function $h$ contribute to the number of parameters/weights. The rest of this section will work toward the proof of Theorem 4.2. The following result states that a $\mathcal{C}$-labelled sequence can be labelled by a hyperplane arrangement classifier of a special form.

**Proposition 4.3.** *Let* $\{(x_i, y_i)\}_{i \in [n]}$ *be* $\mathtt{HAC}(d, r, k)$-*labelled. Then there exist* $\mathbf{V} = [v_1 \cdots v_k] \in \mathbb{R}^{d \times k}, c \in \mathbb{R}^k$ *and* $h \in \mathtt{Bool}_k$ *such that for all* $i \in [n]$, *we have 1)* $y_i = h(\sigma_{\mathtt{sgn}}(\mathbf{V}^\top x_i + c))$, *2)* $\mathrm{rank}(\mathbf{V}) \leq r$ *and 3)* $|v_j^\top x_i + c_j| \geq 1$ *for all* $i \in [n], j \in [k]$.

The proof, given in Appendix A.1, is similar to showing the existence of a max-margin separating hyperplane for a linearly separable dataset.

**Definition 4.4.** *Let* $I$ *be a finite set and let* $a_i \in \mathbb{R}^n$ *for each* $i \in I$. *Let* $A = \{a_i\}_{i \in I}$. *A conical combination of* $A$ *is a linear combination* $\sum_{i \in I} \lambda_i a_i$ *where the weights* $\lambda_i \in \mathbb{R}_{\geq 0}$ *are nonnegative. The conical hull of* $A$, *denoted* $\mathtt{coni}(A)$, *is the set of all conical combinations of* $A$, *i.e.,* $\mathtt{coni}(\{a_i\}_{i \in I}) := \{\sum_{i \in I} \lambda_i a_i : \lambda_i \in \mathbb{R}_{\geq 0}, \forall i \in I\}$.

The result below follows easily from the Carathédory's theorem for the conical hull (Lovász & Plummer, 2009). For the sake of completeness, we included the proof in Appendix A.2.

**Proposition 4.5.** *Let* $a_1, \ldots, a_m \in \mathbb{R}^n$ *and* $b_1, \ldots, b_m \in \mathbb{R}$. *For each subset* $I \subseteq [m]$, *define* $\mathcal{P}_I := \{x \in \mathbb{R}^n : a_i^\top x \leq b_i \forall i \in I\}$. *Suppose that* $\mathcal{P}_{[m]}$ *is nonempty. Then 1)* $\min_{x \in \mathcal{P}_I} \frac{1}{2} \|x\|^2$ *has a unique minimizer, denoted by* $x_I^*$ *below, and 2) there exists a subset* $J \subseteq [m]$ *such that* $|J| = \min\{m, n\}$ *and for all* $I \subseteq [m]$ *with* $J \subseteq I$, *we have* $x_{[m]}^* = x_I^*$.

*Proof of Theorem 4.2.* Let $(x_i, y_i)$ be $\mathtt{HAC}(d, r, k)$-realizable, and $\mathbf{V}, c$ and $h$ be as in Proposition 4.3. For each $i \in [n]$, define the Boolean vectors $s_i := \sigma_{\mathtt{sgn}}(\mathbf{V}^\top x_i + c) \in \{\pm 1\}^k$ and $s_{ij} = \sigma_{\mathtt{sgn}}(v_j^\top x_i + c_j)$ denote the $j$-th entry of $s_i$. Note that $s_{ij}(v_j^\top x_i + c_j) = |v_j^\top x_i + c_j| \geq 1$.

We first outline the steps of the proof:

1. Using a subset of the samples $\{(x_{i_\ell}, y_{i_\ell}) : \ell \in [d(k+1)]\}$ with additional $k(d+1)(1 + \lceil \log_2 k \rceil)$ bits of side information $\{(s_{i_\ell j_\ell}, j_\ell) : \ell \in [d(k+1)]\}$, we can reconstruct $\overline{\mathbf{W}}, \overline{b}$ such that $\sigma_{\mathtt{sgn}}(\overline{\mathbf{W}}^\top x_i + \overline{b}) = s_i$ for all $i \in [n]$.

2. Using an additional subset of samples $\{(x_{\iota_\ell}, y_{\iota_\ell}) : \ell = 1, \ldots, \binom{k}{\leq r}\}$ in conjunction with the $\overline{\mathbf{W}}, \overline{b}$ reconstructed in the previous step, we can find $g \in \mathtt{Bool}_k$ such that $g(s_i) = h(s_i)$ for all $i$.

Now, consider the set

$$\mathcal{P} := \left\{ (\mathbf{W}, b) \in \mathbb{R}^{d \times k} \times \mathbb{R}^k : s_{ij}(w_j^\top x_i + b_j) \geq 1, \forall i \in [n], j \in [k] \right\}.$$

Note that $\mathcal{P}$ is a convex polyhedron in $(d+1)k$-dimensional space. Let $(\overline{\mathbf{W}}, \overline{b})$ be the minimum norm element of $\mathcal{P}$. Note that $\sigma_{\mathtt{sgn}}(\overline{\mathbf{W}}^\top x_i + \overline{b}) = \sigma_{\mathtt{sgn}}(\mathbf{V}^\top x_i + c) = s_i$ by construction.

By Proposition 4.5, there exists a set of tuples

$$\{(i_\ell, j_\ell)\}_{\ell = 1, \ldots, (d+1)k}, \text{ where } (i_\ell, j_\ell) \in [n] \times [k]$$

such that $\overline{\mathbf{W}}, \overline{b}$ is also the minimum norm element of

$$\mathcal{P}' := \left\{ (\mathbf{W}, b) \in \mathbb{R}^{d \times k} \times \mathbb{R}^k : s_{i_\ell j_\ell}(w_{j_\ell}^\top x_{i_\ell} + b_{j_\ell}) \geq 1, \ell = 1, \ldots, d(k+1) \right\}.$$

To encode the defining equations of $\mathcal{P}'$, we need to store

$$\text{samples } \{(x_{i_\ell}, y_{i_\ell})\}_{\ell=1}^{d(k+1)} \text{ and side information } \{(s_{i_\ell j_\ell}, j_\ell)\}_{\ell=1}^{d(k+1)}. \tag{2}$$

Note that each $s_{i_\ell j_\ell}$ requires 1 bit while each $j_\ell \in [k]$ requires $\lceil \log_2 k \rceil$ bits. In total, encoding $\mathcal{P}'$ requires storing $d(k+1)$ samples and $d(k+1)(1 + \lceil \log_2 k \rceil)$ of bits.

To reconstruct $g \in \texttt{Bool}_k$ that agrees with $h$ on all the samples, it suffices to know $h$ when restricted to $\{s_i\}_{i=1}^n$. Since $\{s_i\}_{i=1}^n$ is a subset of $\mathfrak{S}_{\overline{\mathbf{W}}, \bar{b}}$, we have by eq. (1) that $|\{s_i\}_i^n| \le \binom{k}{\le r}$. Thus, $\{s_i\}_{i=1}^n$ has at most $\binom{k}{\le r}$ unique elements. Let $\left\{ s_{\iota_\ell} : \ell = 1, \ldots, \binom{k}{\le r} \right\}$ be a set containing all such unique elements. Thus, we store

$$\text{samples } \{(x_{\iota_\ell}, y_{\iota_\ell}) : \ell = 1, \ldots, \binom{k}{\le r}\}. \tag{3}$$

Using $\overline{\mathbf{W}}, \bar{b}$ as defined above, we have $s_{\iota_\ell} = \sigma_{\texttt{sgn}}(\overline{\mathbf{W}}^\top x_{\iota_\ell} + \bar{b})$. Now, simply choose $g$ such that $g(s_{\iota_\ell}) = y_{\iota_\ell}$ for all $\ell = 1, \ldots, \binom{k}{\le r}$.

To summarize, we formally define the compression and reconstruction functions $(\kappa, \rho)$. Let $\kappa$ take the full sample $\{(x_i, y_i)\}_{i=1}^n$ and output the subsample (and side information) in eq. (2) and eq. (3). The reconstruction function $\rho$ first constructs $\overline{\mathbf{W}}, \bar{b}$ using eq. (2). Next, $\rho$ constructs $g$ using $\overline{\mathbf{W}}, \bar{b}$ and the samples of eq. (3). $\qquad\square$

Now, the following result together with the sample compression scheme for $\texttt{HAC}(d, r, k)$ we constructed imply Theorem 3.6 from the previous section.

**Theorem 4.6** (Littlestone & Warmuth (1986))**.** *If $\mathcal{C}$ has sample compression scheme $(\rho, \kappa)$, then* $\texttt{VC}(\mathcal{C}) \le 8 \cdot \texttt{size}(\rho, \kappa)$.

*Remark* 4.7. Note that the reconstruction function $\rho$ is *not* permutation-invariant. Furthermore, the overall sample compression scheme $\rho, \kappa$ is *not* stable in the sense of Hanneke & Kontorovich (2021). In general, sample compression schemes with permutation-invariant $\rho$ (Floyd & Warmuth, 1995) and *stable* sample compression schemes (Hanneke & Kontorovich, 2021) enjoy tighter generalization bounds compared to ordinary sample compression schemes. We leave as an open question whether $\texttt{HAC}(d, r, k)$ has such specialized compression schemes.

## 5 MINIMAX-OPTIMALITY FOR LEARNING LIPSCHITZ CLASS

In this section, we show that empirical risk minimization (ERM) with respect to the 0-1 loss over $\texttt{HAC}(d, r, k)$, for properly chosen $r$ and $k$, is minimax optimal for classification where the posterior class probability function is $L$-Lipschitz, for fixed $L > 0$. Furthermore, the choices for $r$ and $k$ is such that the associated HANN, the neural network realization of $\texttt{HAC}(d, r, k)$, is overparametrized for the Boolean function implementations discussed in Remark 3.7.

Below, let $X \in [0, 1]^d$ and $Y \in \mathbb{B}$ be the random variables corresponding to a sample and label jointly distributed according to $P$. Write $\eta_P(x) := P(Y = 1 | X = x)$ for the posterior class probability function.

Let $\Sigma(L, [0, 1]^d)$ denote the class of $L$-Lipschitz functions $f : [0, 1]^d \to \mathbb{R}$, i.e.,

$$|f(x) - f(x')| \le L\|x - x'\|_2, \quad \forall x, x' \in [0, 1]^d.$$

The following minimax lower bound result[2] concerns classification when $\eta_P$ is $L$-Lipschitz:

**Theorem 5.1** (Audibert & Tsybakov (2007))**.** *There exists a constant $C > 0$ such that*

$$\inf_{\tilde{f}_n} \sup_{P \,:\, \eta_P \in \Sigma(L, [0,1]^d)} \mathbb{E}[R(\tilde{f}_n)] - R_P^* \ge Cn^{-\frac{1}{d+2}}.$$

The infimum above is taken over all possible learning algorithms $\tilde{f}_n$, i.e., mappings from $(\mathcal{X} \times \mathbb{B})^n$ to Borel measurable functions $\mathcal{X} \to \mathbb{B}$. When $\hat{f}_n$ is an empirical risk minimizer (ERM) over $\texttt{HAC}(d, r, k)$ where $d = r$ for $k = n^{\frac{1}{d+2}}$, this minimax rate is achieved.

---

[2]The result we cite here is a special case of (Audibert & Tsybakov, 2007, Theorem 3.5), which gives minimax lower bound for when $\eta_P$ has additional smoothness assumptions.

**Theorem 5.2.** *Let $d \geq 1$ be fixed. Let $\hat{f}_n$ be an ERM over* `HAC`$(d, d, k)$ *where* $k = k(n) \asymp n^{\frac{1}{d+1}}$. *Then there exists a constant $C'$ such that*

$$\sup_{P \,:\, \eta_P \in \Sigma(L, [0,1]^d)} \mathbb{E}[R(\hat{f}_n)] - R_P^* \leq C' n^{-\frac{1}{d+2}}.$$

*Proof sketch (see Appendix A.3 for full proof).* We first show that the histogram classifier over the standard partition of $[0, 1]^d$ into smaller cubes is an element of $\mathcal{C} :=$ `HAC`$(d, d, k)$, thus reducing the problem to proving minimax-optimality of the histogram classifier. Previous work Györfi et al. (2006) Theorem 4.3 established this for the histogram *regressor*. The analogous result for the histogram *classifier*, to the best of our knowledge, has not appeared in the literature and thus is included for completeness.

The neural network implementation of `HAC`$(d, d, k)$ where $k \asymp n^{1/(d+2)}$ in Theorem 5.2 can be overparametrized. Using either the 1- or the 2-hidden layer neural network implementations of Boolean functions as in Remark 3.7, the resulting HANN is overparametrized and has number of weights either $\geq k2^k$ or $\geq c^2 2^k/k$ respectively. Both $k2^k$ and $c^2 2^k/k \gg n$ exponentially while `VC`(`HAC`$(d, d, k)$) $= o(n)$.

## 6 EMPIRICAL RESULTS

In this section, we discuss experimental results of using HANNs for classifying synthetic and real datasets. Our implementation uses TensorFlow (Abadi et al., 2016) with the Larq (Geiger & Team, 2020) library for training neural networks with threshold activations. Note that Theorem 5.2 holds for ERM with respect to the 0-1 loss over HANNs, which is intractable in practice. Furthermore, our theory is for binary classification, while some of the datasets in the experiments are multiclass.

**Synthetic datasets.** We apply a HANN (model specification shown in Figure 3-top left) to the MOONS synthetic dataset with two classes with the hinge loss.

The heuristic for training networks with threshold activation can significantly affect the performance (Kim et al., 2019). We consider two of the most popular heuristics: the straight-through-estimator (SteSign) and the SwishSign, introduced by Hubara et al. (2017) and Darabi et al. (2019), respectively. Below, we use SwishSign since it reliably leads to higher validation accuracy (Figure 3-bottom left), consistent with the finding of Darabi et al. (2019).

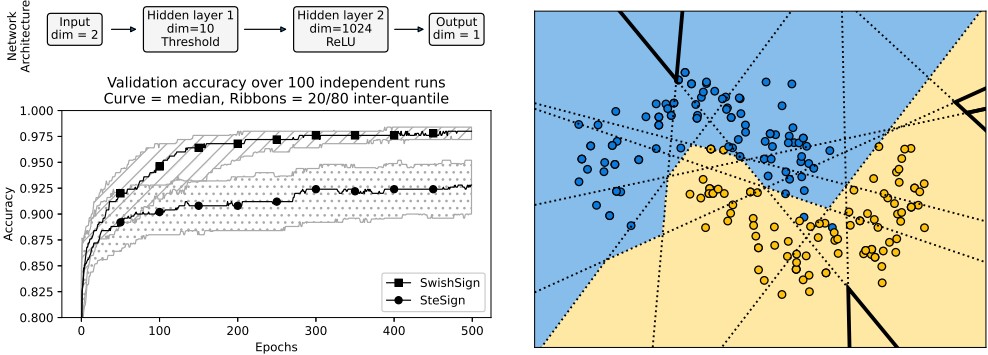

Figure 3: *Top left.* Architecture of HANN used for the MOONS dataset (`make_moons` in `sklearn`). *Bottom left.* Validation accuracies from 100 independent runs with random initialization and data. *Right.* Dotted lines denote the hyperplane arrangement. Coloring of the cells denote the decision region of the trained classifier. A cell $\Delta$ is bolded if 1) no training data lies in $\Delta$ and 2) $\Delta$ does not touch the decision boundary.

The width of the hidden layer is $2^{10} = 1024$. Thus, by Mukherjee & Basu (2017) Lemma 2.1, the model can assign labels to the bolded cells arbitrarily without changing the training loss. Nevertheless, the optimization appears to be biased toward a topologically simpler classifier. This behavior is consistently reproducible. See Figure 7.

**Real-world datasets.** Klambauer et al. (2017) introduced *self-normalizing neural networks* (SNN) which were shown to outperform other neural networks on a panel of 121 UCI datasets. Subsequently, Wu et al. (2018) proposed the *dendritic neural network* architecture, which further improved classification performance on this panel of datasets. Following their works, we evaluate the performance of HANNs on the 121 UCI datasets.

A crucial hyperparameter for HANN is the number of hyperplanes $k$. We ran the experiments with $k \in \{15, 100\}$ to test the impact on accuracy. The Boolean function $h$ is implemented as a 1-hidden layer residual network (He et al., 2016) of width 1000. The logistic loss is used.

We use the same train, validation, and test sets as in Klambauer et al. (2017). The reported accuracies on the held-out test set are based on the model with the highest validation accuracy. The models will be referred to as HANN15 and HANN100, respectively. The results are shown in Figure 4. The accuracies of SNN and DENN are obtained from Table A1 in the supplemental materials of Wu et al. (2018). Full details for the training and accuracy tables can be found in the appendix.

Comparing HANN with SNN and DENN over the UCI datasets

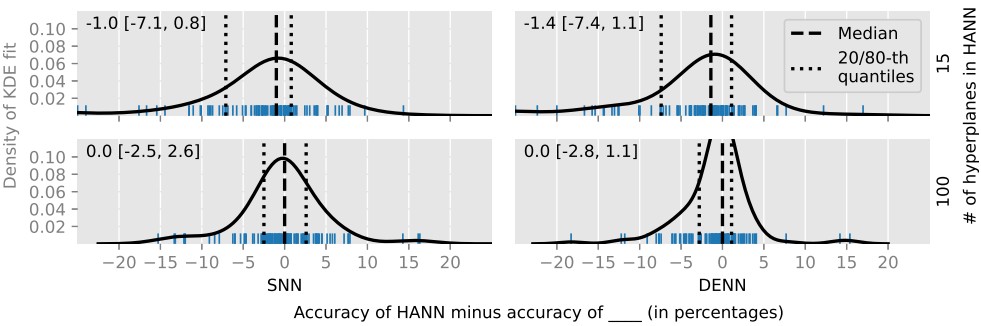

Figure 4: Each blue tick above the x-axis represents a single dataset, where the x-coordinate of the tick is the difference of the accuracy of HANN and either SNN (left) or DENN (right) on the dataset. The solid black curves are kernel density estimates for the blue ticks. The number of hyperplanes used by HANN is either 15 (top) or 100 (bottom). The quantities shown in the top-left corner of each subplot are the median, 20-th and 80-th quantiles of the differences, respectively, rounded to 1 decimal place.

The HANN15 model (top row of Figure 4) already achieves median accuracy within 1.5% of both SNN and DENN. With the larger HANN100 model (bottom row), the gap is reduced to zero. The largest training set in this panel of datasets has size 77904. The HANN15 and HANN100 models use $\approx 10^4$ and $10^5$ weights, respectively. By comparison, the average numbers of weights used by SNN and DENN are both $\geq 5 * 10^5$. Details on these estimates are included in Appendix C. Thus, all three models considered here are overparametrized.

## 7 DISCUSSION

We have introduced an architecture for which the VC theorem can be used to prove minimax-optimality of ERM over HANNs in an overparametrized setting with Lipschitz posterior. To our knowledge, this is the first time VC theory has been used to analyze the performance of a neural network in the overparametrized regime. Furthermore, the same architecture leads to state-of-the-art performance over a benchmark collection of unstructured datasets.

To the best of our knowledge, no existing theoretical bound for overparametrized NNs yields meaningful results. Yet there is immense interest in understanding what aspects of deep NNs can explain their performance, even if the bounds aren't yet small (Bartlett et al., 2017; Neyshabur et al., 2017; Jiang et al., 2019). Our work shows that the compressibility of the network, as reflected by the sample compression scheme, is a useful avenue, and one that has not previously been explored – ours is the first work applying sample compression to NNs. This seems likely to open the door to further analysis of quantized NNs.

ACKNOWLEDGEMENTS

The authors were supported in part by the National Science Foundation under awards 1838179 and 2008074, and by the Department of Defense, Defense Threat Reduction Agency under award HDTRA1-20-2-0002.

REPRODUCIBILITY STATEMENT

All code for downloading and parsing the data, training the models, and generating plots in this manuscript are available at `https://github.com/YutongWangUMich/HANN`. Complete proofs for all novel results are included in the main article or in an appendix.

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

## A PROOFS

### A.1 PROOF OF PROPOSITION 4.3

By definition, there exists $h \in \texttt{Bool}_k$, $\mathbf{W} \in \mathbb{R}^{d \times k}$ of rank at most $r$, and $b \in \mathbb{R}^k$ such that $y_i = h(\sigma_{\texttt{sgn}}(\mathbf{W}^\top x_i + b))$.

Now, let $j \in [k]$ be fixed. Since $|w_j^\top x_i + b_j| \geq 0$ for all $i \in [n]$, there exists a small perturbation $\tilde{c}_j$ of $b_j$ such that $|w_j^\top x_i + \tilde{c}_j| > 0$ for all $i \in [n]$. Now, let $\lambda_j := \min_{i \in [n]} |w_j^\top x_i + \tilde{c}_j|$ which is positive. Define $v_j := w_j/\lambda_j$ and $c_j = \tilde{c}_j/\lambda_j$, we have $|v_j^\top x_i + c_j| \geq 1$ for all $i \in [n]$, as desired. Note that $\text{rank}(\mathbf{V}) = \text{rank}(\mathbf{W})$. □

### A.2 PROOF OF PROPOSITION 4.5

If $m \leq n$, then the result holds trivially by letting $J = [m]$. Below, suppose that $m > n$. Let $g_i(x) = a_i^\top x - b_i$ for each $i \in [m]$ and $f(x) = \frac{1}{2}\|x\|_2^2$. Then $\nabla f(x) = x$ and $\nabla g_i(x) = a_i$. By definition, $x_I^*$ is a minimizer of

$$\min_{x \in \mathbb{R}^n} f(x) \text{ s.t. } g_i(x) \leq 0, \, \forall i \in I,$$

which is a convex optimization with strongly convex objective. Thus, the minimizer $x_I^*$ is unique and furthermore is the unique element $x$ of $\mathbb{R}^n$ satisfying the KKT conditions:

$$x \in \mathcal{P}_I \text{ and } \exists \text{ a set of nonnegative weights } \{\lambda_i\}_{i \in I} \text{ such that } -x = \sum_{i \in I} \lambda_i a_i.$$

Thus, $x_I^*$ can be equivalently characterized as the unique element of $x \in \mathbb{R}^n$ satisfying

$$x \in \mathcal{P}_I \text{ and } -x \in \texttt{coni}(\{a_i\}_{i \in I}). \tag{4}$$

In particular, $x_{[m]}^* \in \mathcal{P}_{[m]}$ and $-x_{[m]}^* \in \texttt{coni}(\{a_i\}_{i \in [m]})$. By the Carathédory's theorem for the conical hull (Lovász & Plummer, 2009), there exists $\underline{I} \subseteq [m]$ such that $|\underline{I}| = n$ and $-x_{[m]}^* \in \texttt{coni}(\{a_i\}_{i \in \underline{I}})$. Thus, for any $J \subseteq [m]$ such that $\underline{I} \subseteq J$, we have $-x_{[m]}^* \in \texttt{coni}(\{a_i\}_{i \in J})$. Furthermore, $J \subseteq [m]$ implies $\mathcal{P}_J \supseteq \mathcal{P}_{[m]}$. In particular, $x_{[m]}^* \in \mathcal{P}_J$. Putting it all together, we have $x_{[m]}^* \in \mathcal{P}_J$ and $-x_{[m]}^* \in \texttt{coni}(\{a_i\}_{i \in J})$. By the uniqueness, we have $x_J^* = x_{[m]}^*$. $\qquad\square$

### A.3 PROOF OF THEOREM 5.2

In this proof, the constant $C$ does not depending on $n$, and may change from line to line.

We fix a joint distribution $P$ such that $\eta_P \in \Sigma(L, [0,1]^d)$ throughout the proof. Thus, the notation for risks will omit the $P$ in their subscript, e.g., we write $\hat{R}_n(f)$ instead of $\hat{R}_{P,n}(f)$ and $R^*$ instead of $R_P^*$. Below, let $\beta > \alpha > 0$ be constants such that $\alpha d n^{1/(d+2)} \le k \le \beta d n^{1/(d+2)}$. Let $\tilde{k} := \lceil k/d \rceil$.

Let $\mathcal{R}_1, \mathcal{R}_2, \dots, \mathcal{R}_{\tilde{k}^d}$ denote the hypercubes of side length $\ell = 1/\tilde{k}$ forming a partition of $[0,1]^d$. For each $i \in [\tilde{k}^d]$, let $\mathcal{R}_i^- := \{x \in \mathcal{R}_i : \eta_P(x) < 1/2\}$ and $\mathcal{R}_i^+ := \{x \in \mathcal{R}_i : \eta_P(x) \ge 1/2\}$.

Let $\tilde{f} : [0,1]^d \to \mathbb{B}$ be the classifier such that

$$\tilde{f}(x) = \begin{cases} +1 & : x \in \mathcal{R}_i, \int_{\mathcal{R}_i} \eta_P(x) dP(x) \ge \int_{\mathcal{R}_i} (1 - \eta_P(x)) dP(x) \\ -1 & : x \in \mathcal{R}_i, \int_{\mathcal{R}_i} \eta_P(x) dP(x) < \int_{\mathcal{R}_i} (1 - \eta_P(x)) dP(x). \end{cases}$$

In other words, $\tilde{f}$ classifies all $x \in \mathcal{R}_i$ as $+1$ if and only if $P(Y = 1 | X \in \mathcal{R}_i) \ge 1/2$. This is commonly referred to as the *histogram classifier* (Györfi et al., 2006). It is easy to see that

$$P(\tilde{f}(X) \ne Y, X \in \mathcal{R}_i) = \min\left\{ \int_{\mathcal{R}_i} (1 - \eta_P(x)) dP(x), \int_{\mathcal{R}_i} \eta_P(x) dP(x) \right\}$$

For the remainder of this proof, we write "$\sum_i$" to mean "$\sum_{i \in [\tilde{k}^d]}$". Thus,

$$R(\tilde{f}) = \sum_i P(\tilde{f}(X) \ne Y, X \in \mathcal{R}_i) = \sum_i \min\left\{ \int_{\mathcal{R}_i} (1 - \eta_P(x)) dP(x), \int_{\mathcal{R}_i} \eta_P(x) dP(x) \right\}.$$

Next, we note that $\tilde{f} \in \texttt{HAC}(d, d, k)$. To see this, let $j \in [d]$. Take $H_{j1}, \dots, H_{j(\tilde{k}-1)} \subseteq \mathbb{R}^d$ to be the hyperplanes perpendicular to the $j$-th coordinate where, for each $\ell \in [\tilde{k}]$, $H_{j\ell}$ intersects the $j$-th coordinate axis at $\ell/\tilde{k}$. Consider the hyperplane arrangement consisting of all $\{H_{j\ell}\}_{j \in [d], \ell \in [\tilde{k}-1]}$ and let $\{C_1, C_2, \dots\}$ be its cells. Then $\{C_1 \cap [0,1]^d, C_2 \cap [0,1]^d, \dots\} = \{\mathcal{R}_1, \dots, \mathcal{R}_{\tilde{k}^d}\}$ is the partition of $[0,1]^d$ by $1/\tilde{k}$ side length hypercubes. See Figure 5.

Let $\mathbf{W}$ be the matrix of normal vectors and $b$ be the vector of offsets representing this hyperplane arrangement, which requires $d(\tilde{k}-1) = d(\lceil k/d \rceil - 1) \le d(k/d) = k$ hyperplanes. Since $\tilde{f}$ is constant on $\mathcal{R}_i$, there exists a Boolean function $h \in \texttt{Bool}_k$ such that $h \circ q_{\mathbf{W},b}|_{[0,1]^d} = \tilde{f}$. From this, we conclude that $\tilde{f} \in \texttt{HAC}(d, d, k)$.

Thus $\hat{R}_n(\hat{f}_n) - \hat{R}_n(\tilde{f}) \le 0$ and so

$$R(\hat{f}_n) - R^* = R(\hat{f}_n) - \hat{R}_n(\hat{f}_n) + \underbrace{\hat{R}_n(\hat{f}_n) - \hat{R}_n(\tilde{f})}_{\le 0} + \hat{R}_n(\tilde{f}) - R(\tilde{f}) + R(\tilde{f}) - R^*$$

$$\le \underbrace{R(\hat{f}_n) - \hat{R}_n(\hat{f}_n)}_{\text{Term 1}} + \underbrace{\hat{R}_n(\tilde{f}) - R(\tilde{f})}_{\text{Term 2}} + \underbrace{R(\tilde{f}) - R^*}_{\text{Term 3}}.$$

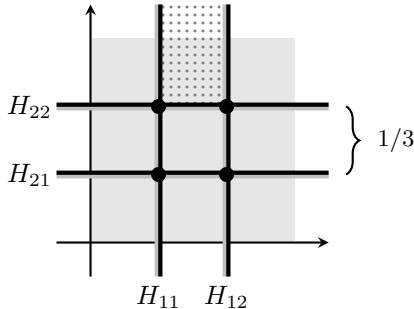

Figure 5: Partition of $[0,1]^d$ into $1/\tilde{k}$ hypercubes via arrangement of $d(\tilde{k}-1)$ hyperplanes, where $d = 2$ and $\tilde{k} = 3$. Shaded region is $[0,1]^d$. Dotted region is a cell of the hyperplane arrangement.

We now bound Terms 1 and 2 using the uniform deviation bound. From Theorem 3.6, we know that there exists a constant $C$ independent of $n$ such that

$$\text{VC}(\text{HAC}(d,d,k)) \leq 8 \cdot \left( k(d+1) + k(d+1)(1 + \lceil \log_2(k) \rceil) + \binom{k}{\leq d} \right) \leq Ck^d.$$

Thus, by Theorem 3.5 with $\delta = 1/(2n)$ and a union bound, with probability at least $1 - 1/n$

$$\max\left\{ |\hat{R}_n(\hat{f}_n) - R(\hat{f}_n)|, |\hat{R}_n(\tilde{f}) - R(\tilde{f})| \right\} \leq C\sqrt{\frac{k^d + \log(n)}{n}} \tag{5}$$

for some $C > 0$.

Next, we focus on Term 3. Recall that

$$R^* = \int_{[0,1]^d} \min\{\eta_P(x), 1 - \eta_P(x)\} dP(x) = \sum_i \int_{\mathcal{R}_i} \min\{\eta_P(x), 1 - \eta_P(x)\} dP(x)$$

and that

$$R(\tilde{f}) = \sum_i \min\left\{ \int_{\mathcal{R}_i} \eta_P(x) dP(x), \int_{\mathcal{R}_i} 1 - \eta_P(x) dP(x) \right\}.$$

Fix some $i \in [k^d]$. Our goal now is to bound the difference between the $i$-th summands in the above expressions for $R(\tilde{f})$ and $R^*$:

$$\min\left\{ \int_{\mathcal{R}_i} \eta_P(x) dP(x), \int_{\mathcal{R}_i} 1 - \eta_P(x) dP(x) \right\} - \int_{\mathcal{R}_i} \min\{\eta_P(x), 1 - \eta_P(x)\} dP(x). \tag{6}$$

First, consider the case that

$$\min\left\{ \int_{\mathcal{R}_i} \eta_P(x) dP(x), \int_{\mathcal{R}_i} 1 - \eta_P(x) dP(x) \right\} = \int_{\mathcal{R}_i} \eta_P(x) dP(x). \tag{7}$$

We claim that there must exist $x_0 \in \mathcal{R}_i$ such that $\eta_P(x_0) \leq 1/2$. Suppose $\eta_P(x) > 1/2$ for all $x \in \mathcal{R}_i$. Then $\eta_P(x) > 1/2 > 1 - \eta_P(x)$. Since $\eta_P(x)$ is continuous, this would contradict eq. (7).

Continue assuming eq. (7), we further divide into two subcases: (1) $\eta_P(x) \leq 1/2$ for all $x \in \mathcal{R}_i$, and (2) there exists some $x_1 \in \mathcal{R}_i$ such that $\eta_P(x_1) > 1/2$.

Under subcase (1), $\min\{\eta_P(x), 1 - \eta_P(x)\} = \eta_P(x)$ for all $x \in \mathcal{R}_i$ in which case $eq.$ (6) $= 0$.

Under subcase (2), since $\eta_P(x_0) \leq 1/2 < \eta_P(x_1)$, we know by the intermediate value theorem that there must exist $x' \in \mathcal{R}_i$ such that $\eta_P(x') = 1/2$. Now,

$$
\begin{aligned}
eq.\ (6) &= \int_{\mathcal{R}_i} (\eta_P(x) - \min\{\eta_P(x), 1 - \eta_P(x)\})dP(x) \\
&\leq \int_{\mathcal{R}_i^+} (\eta_P(x) - \min\{\eta_P(x), 1 - \eta_P(x)\})dP(x) \\
&\quad + \int_{\mathcal{R}_i^-} (\eta_P(x) - \min\{\eta_P(x), 1 - \eta_P(x)\})dP(x) \\
&= \int_{\mathcal{R}_i^+} (\eta_P(x) - (1 - \eta_P(x))dP(x) \qquad \because \text{Definition of } \mathcal{R}_i^{\pm} \\
&\quad + \int_{\mathcal{R}_i^-} (\eta_P(x) - \eta_P(x))dP(x) \\
&= \int_{\mathcal{R}_i^+} (2\eta_P(x) - 1)dP(x) \\
&= 2\int_{\mathcal{R}_i^+} (\eta_P(x) - \eta_P(x'))dP(x) \qquad \because 2\eta_P(x') = 1 \\
&\leq 2L \int_{\mathcal{R}_i^+} \|x - x'\|_2 dP(x) \\
&\leq 2L\sqrt{d} \Pr(\mathcal{R}_i)/\tilde{k} \qquad \because \|x - x'\|_2 \leq \sqrt{d}\|x - x'\|_1 \leq \sqrt{d}(1/\tilde{k}) \\
&\leq 2Ld^{3/2} \Pr(\mathcal{R}_i)/k \qquad \because 1/\tilde{k} = 1/\lceil k/d \rceil \leq 1/(k/d) = d/k.
\end{aligned}
$$

Thus, under assumption eq. (7), we have proven that $eq.\ (6) \leq 2Ld^{3/2}/k$. For the other assumption, i.e., the minimum in eq. (7) is attained by $\int_{\mathcal{R}_i} 1 - \eta_P(x)dP(x)$, a completely analogous argument again shows that $eq.\ (6) \leq 2Ld^{3/2}/k$.

Putting it all together, we have

$$
R(\tilde{f}_n) - R^* \leq 2Ld^{3/2} \sum_i P(\mathcal{R}_i)/k = 2Ld^{3/2}/k. \tag{8}
$$

We have shown that, with probability at least $1 - 1/n$,

$$
R(\hat{f}_n) - R^* \leq C\sqrt{\frac{k^d + \log(n)}{n}} + \frac{2Ld^{3/2}}{k}.
$$

Using $\alpha d n^{1/(d+2)} \leq k \leq \beta d n^{1/(d+2)}$, we have with probably at least $1 - 1/n$ that

$$
\begin{aligned}
R(\hat{f}_n) - R^* &\leq C\sqrt{\frac{k^d + \log(n)}{n}} + \frac{2Ld^{3/2}}{k} \\
&\leq C\sqrt{\frac{(\beta d)^d n^{d/(d+2)} + \log(n)}{n}} + \frac{2Ld^{3/2}}{\alpha d n^{1/(d+2)}} \\
&\leq C\left(\sqrt{\frac{n^{d/(d+2)}}{n}} + n^{-1/(d+2)}\right) \qquad \because \log(n) = o(n^{1/d+2}) \\
&= C\left(\sqrt{n^{-2/(d+2)}} + n^{-1/(d+2)}\right) \\
&\leq Cn^{-\frac{1}{d+2}}.
\end{aligned}
$$

Taking expectation, we have $\mathbb{E}[R(\hat{f}_n)] - R^* \leq (1 - 1/n)Cn^{-\frac{1}{d+2}} + 1/n \cdot 1 \leq Cn^{-\frac{1}{d+2}}$. $\qquad \square$

## B  TRAINING DETAILS

**Data preprocessing.** The pooled training and validation data is centered and standardized using the `StandardScaler` function from sklearn. The transformation is also applied to the test data, using the centers and scaling from the pooled training and validation data:

```
scaler = StandardScaler().fit(X_train_valid)
X_train_valid = scaler.transform(X_train_valid)
X_test = scaler.transform(X_test)
```

If the feature dimension and training sample size are both $> 50$, then the data is dimension reduced to 50 principal component features:

```
if min(X_train_valid.shape) > 50:
    pca = PCA(n_components = 50).fit(X_train_valid)
    X_train_valid = pca.transform(X_train_valid)
    X_test = pca.transform(X_test)
```

Note that this is equivalent to freezing the weights between the Input and the Latent layer in Figure 2.

**Validation and test accuracy.** Every 10 epochs, the validation accuracy during the past 10 epochs are averaged. A smoothed validation accuracy is calculated as follows:

```
val_acc_sm = (1-sm_param)*val_acc_sm + sm_param*val_acc_av
## Variable description:
# sm_param = 0.1
# val_acc_av = average of the validation in the past 10 epochs
# val_acc_sm = smoothed validation accuracy
```

The predicted test labels is based on the snapshot of the model at the highest smoothed validation accuracy, at the end once max epochs is reached.

**Heuristic for coarse gradient of the threshold function.** We use the SwishSign from the Larq library (Geiger & Team, 2020).

```
# import larq as lq
qtz = lq.quantizers.SwishSign()
```

**Dropout.** During training, dropout is applied to the Boolean output of the threshold function, i.e, the variables $B_1, B_2, \ldots, B_k$ in Figure 2. This improves generalization by preventing the training accuracy from reaching $100\%$.

```
# from tensorflow.keras.layers import Dense, Dropout
hyperplane_enc = Dense(n_hyperplanes, activation = qtz)(inputs)
hyperplane_enc = Dropout(dropout_rate)(hyperplane_enc)
```

**Implementation of the Boolean function.** For the Boolean function $h$, we use a 1-hidden layer residual network (He et al., 2016) with 1000 hidden nodes:

```
# from tensorflow.keras.layers import Dense, Add
# output_dim = num_classes
n_hidden = 1000
hidden = Dense(n_hidden, activation="relu")(hyperplane_enc)
out_hidden = Dense(output_dim, activation = "linear")(hidden)
out_skip = Dense(output_dim, activation = "linear")(hyperplane_enc)
outputs = Add()([out_skip,out_hidden])
```

**Hyperparameters.** HANN15 is trained with a hyperparameter grid of size 3 where only the dropout rate is tuned. The hyperparameters are summarized in Table 2. The model with the highest smoothed validation accuracy is chosen.

The model HANN15 is trained with the following hyperparameters:

Table 1: `HANN15` model and training hyperparameter grid

| OPTIMIZER | SGD |
|---|---|
| LEARNING RATE | 0.01 |
| DROPOUT RATE | $\{0.1, 0.25, 0.5\}$ |
| MINIBATCH SIZE | 128 |
| BOOLEAN FUNCTION | 1-HIDDEN LAYER RESNET WITH 1000 HIDDEN NODES |
| EPOCHS | 100 MINIBOONE 5000 FOR ALL OTHERS |

For `HANN100`, we only used 1 set of hyperparameters.

Table 2: `HANN100` model and training hyperparameter

| OPTIMIZER | SGD |
|---|---|
| LEARNING RATE | 0.01 |
| DROPOUT RATE | 0.5 |
| MINIBATCH SIZE | 128 |
| BOOLEAN FUNCTION | 1-HIDDEN LAYER RESNET WITH 1000 HIDDEN NODES |
| EPOCHS | 100 MINIBOONE 5000 FOR ALL OTHERS |

## C  PARAMETER COUNTS

The widest part of `HANN15` and `HANN100` models are the weights mapping from $\mathbb{B}^k$ ($k$ = number of hyperplanes) to $\mathbb{R}^{1000}$ (1000 = number of hidden layer of the boolean function) where $k \in \{15, 100\}$. Thus, the two HANN models use $\geq 15 \times 1000 \geq 10^4$ and $\geq 100 \times 1000 = 10^5$ weights, respectively.

The weight count estimates for the Self-normalized Neural Network (SNN) and Dendritic Neural Network (DENN) use the formula $(\# \text{ layers } - 1) \times (\# \text{ neurons per layer})^2$.

For the Self-normalized Neural Network (SNN), average number of layers = 10.8, and the number of neurons per layers $\geq 256$, found on page 7 and Table A4 of Klambauer et al. (2017), respectively. The number of weights is $\geq (10 - 1) * (256^2) = 655,360$ weights.

The parameters for the dendritic neural network (DENN) is found in the public GitHub repository xiangwenliu/DENN of Wu et al. (2018) which lists number of layers = 3 and number of neurons per layer = 512, found on line 41 and 52 of `train_uci.py`, respectively. The number of weights is $\geq (3 - 1) * (512^2) = 524,288$ weights.

## D  ADDITIONAL PLOTS

**Multiclass hinge versus cross-entropy loss.** Figure 6 shows the accuracy differences when the Weston-Watkins hinge loss is used. Compared to the results shown in Figure 4, the performance for `HANN100` is slightly worse and the performance for `HANN15` is slightly better.

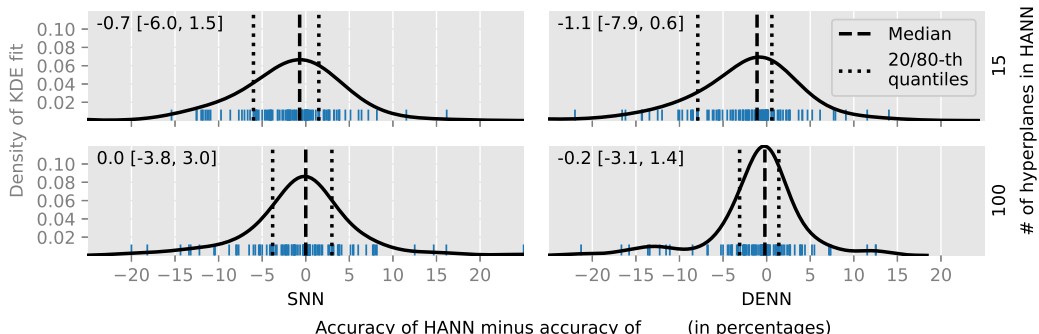

Figure 6: Each blue tick above the x-axis represents a single dataset, where the x-coordinate of the tick is the difference of the accuracy of HANN and either SNN (left) or DENN (right) on the dataset. The number of hyperplanes used by HANN is either 15 (top) or 100 (bottom). The quantities shown in the top-left corner of each subplot are the median, 20-th and 80-th quantiles of the differences, respectively, rounded to 1 decimal place.

**Implicit bias for low complexity decision boundary.** In Figure 7, we show additional results ran with the same setting for the MOONS synthetic dataset as in the Empirical Results section. From the perspective of the training loss, the label assignment in the bold-boundary regions is irrelevant. Nevertheless, the optimization consistently appears to be biased toward the geometrically simpler classifier, despite the capacity for fitting complex classifiers.

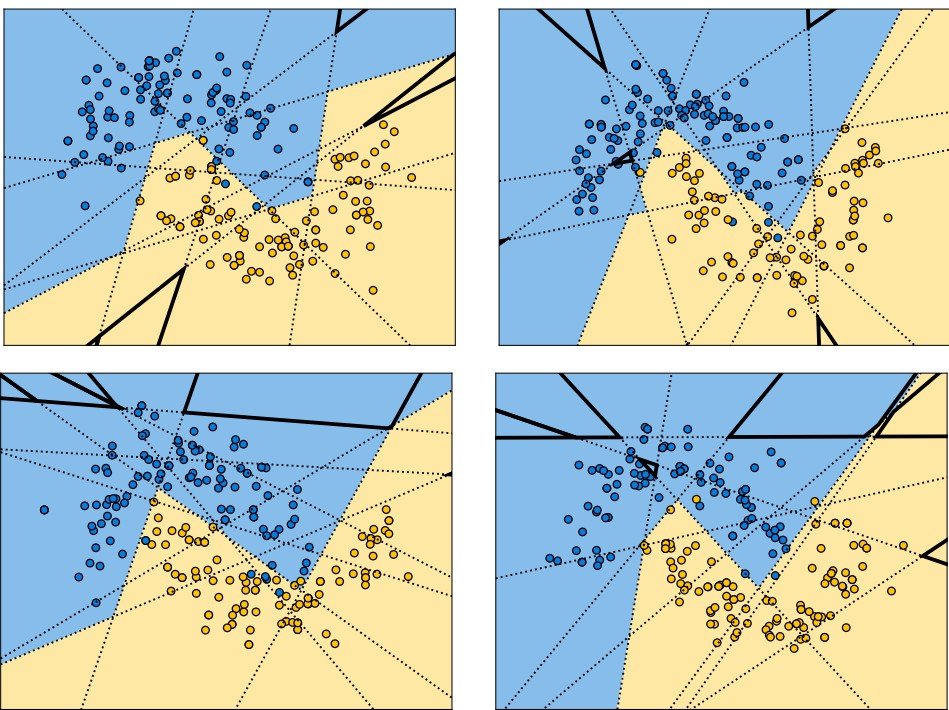

Figure 7: Four independent runs of HANN on the MOONS synthetic dataset. Similar to Figure 3.

## E  TABLE OF ACCURACIES

Below is the table of accuracies used to make Figure 4. The last column "HANN100trn" records the *training* accuracy at the epoch of the highest validation accuracy.

| DSName | HANN15 | HANN100 | SNN | DENN | HANN100trn |
|---|---|---|---|---|---|
| abalone | 63.41 | 65.13 | 66.57 | 66.38 | 70.60 |
| acute-inflammation | 100.00 | 100.00 | 100.00 | 100.00 | 100.00 |
| acute-nephritis | 100.00 | 100.00 | 100.00 | 100.00 | 100.00 |
| adult | 84.32 | 85.04 | 84.76 | 84.80 | 86.35 |
| annealing | 47.00 | 74.00 | 76.00 | 75.00 | 96.81 |
| arrhythmia | 62.83 | 64.60 | 65.49 | 67.26 | 97.19 |
| audiology-std | 56.00 | 68.00 | 80.00 | 76.00 | 99.79 |
| balance-scale | 92.95 | 96.79 | 92.31 | 98.08 | 98.32 |
| balloons | 100.00 | 100.00 | 100.00 | 100.00 | 100.00 |
| bank | 88.50 | 88.05 | 89.03 | 89.65 | 95.62 |
| blood | 75.94 | 75.40 | 77.01 | 73.26 | 82.03 |
| breast-cancer | 70.42 | 63.38 | 71.83 | 69.01 | 86.63 |
| breast-cancer-wisc | 97.71 | 98.29 | 97.14 | 97.71 | 98.65 |
| breast-cancer-wisc-diag | 97.89 | 98.59 | 97.89 | 98.59 | 99.65 |
| breast-cancer-wisc-prog | 73.47 | 71.43 | 67.35 | 71.43 | 98.50 |
| breast-tissue | 61.54 | 80.77 | 73.08 | 65.38 | 95.78 |
| car | 98.84 | 100.00 | 98.38 | 98.84 | 99.32 |
| cardiotocography-10clases | 78.91 | 82.11 | 83.99 | 82.30 | 94.69 |
| cardiotocography-3clases | 90.58 | 93.97 | 91.53 | 94.35 | 97.09 |
| chess-krvk | 47.75 | 72.77 | 88.05 | 80.41 | 67.16 |
| chess-krvkp | 98.62 | 99.37 | 99.12 | 99.62 | 99.93 |
| congressional-voting | 61.47 | 57.80 | 61.47 | 57.80 | 66.54 |
| conn-bench-sonar-mines-rocks | 78.85 | 84.62 | 78.85 | 82.69 | 99.51 |
| conn-bench-vowel-deterding | 89.39 | 98.92 | 99.57 | 99.35 | 98.38 |
| connect-4 | 78.96 | 86.39 | 88.07 | 86.46 | 86.83 |
| contrac | 52.72 | 49.73 | 51.90 | 54.89 | 62.95 |
| credit-approval | 81.98 | 79.65 | 84.30 | 82.56 | 98.47 |
| cylinder-bands | 69.53 | 73.44 | 72.66 | 78.12 | 99.54 |
| dermatology | 98.90 | 97.80 | 92.31 | 97.80 | 99.63 |
| echocardiogram | 84.85 | 87.88 | 81.82 | 87.88 | 94.62 |
| ecoli | 86.90 | 84.52 | 89.29 | 85.71 | 92.83 |
| energy-y1 | 93.23 | 97.40 | 95.83 | 95.83 | 97.84 |
| energy-y2 | 89.06 | 91.15 | 90.63 | 90.62 | 95.72 |
| fertility | 92.00 | 92.00 | 92.00 | 88.00 | 95.44 |
| flags | 39.58 | 50.00 | 45.83 | 52.08 | 94.96 |
| glass | 77.36 | 60.38 | 73.58 | 60.38 | 96.77 |
| haberman-survival | 72.37 | 65.79 | 73.68 | 65.79 | 82.34 |
| hayes-roth | 71.43 | 82.14 | 67.86 | 85.71 | 85.61 |
| heart-cleveland | 53.95 | 59.21 | 61.84 | 57.89 | 96.65 |
| heart-hungarian | 72.60 | 79.45 | 79.45 | 78.08 | 95.67 |
| heart-switzerland | 45.16 | 51.61 | 35.48 | 48.39 | 89.86 |
| heart-va | 36.00 | 30.00 | 36.00 | 32.00 | 94.34 |
| hepatitis | 82.05 | 82.05 | 76.92 | 79.49 | 100.00 |
| hill-valley | 66.83 | 68.81 | 52.48 | 54.62 | 72.69 |
| horse-colic | 80.88 | 83.82 | 80.88 | 82.35 | 97.65 |
| ilpd-indian-liver | 70.55 | 69.18 | 69.86 | 71.92 | 86.75 |
| image-segmentation | 87.76 | 90.57 | 91.14 | 90.57 | 99.31 |
| ionosphere | 89.77 | 87.50 | 88.64 | 96.59 | 98.84 |
| iris | 100.00 | 97.30 | 97.30 | 100.00 | 97.39 |
| led-display | 73.60 | 75.20 | 76.40 | 76.00 | 75.96 |

Continued on next page

| DSName | HANN15 | HANN100 | SNN | DENN | HANN100trn |
|---|---|---|---|---|---|
| lenses | 50.00 | 66.67 | 66.67 | 66.67 | 100.00 |
| letter | 81.82 | 96.86 | 97.26 | 96.20 | 96.47 |
| libras | 64.44 | 81.11 | 78.89 | 77.78 | 98.20 |
| low-res-spect | 86.47 | 90.23 | 85.71 | 90.23 | 99.81 |
| lung-cancer | 37.50 | 62.50 | 62.50 | 62.50 | 100.00 |
| lymphography | 89.19 | 94.59 | 91.89 | 94.59 | 99.67 |
| magic | 86.52 | 87.49 | 86.92 | 86.81 | 87.84 |
| mammographic | 81.25 | 80.00 | 82.50 | 80.83 | 87.10 |
| miniboone | 90.04 | 90.73 | 93.07 | 93.30 | 89.98 |
| molec-biol-promoter | 73.08 | 80.77 | 84.62 | 88.46 | 99.38 |
| molec-biol-splice | 79.05 | 78.04 | 90.09 | 85.45 | 98.35 |
| monks-1 | 65.97 | 69.91 | 75.23 | 81.71 | 99.89 |
| monks-2 | 66.20 | 66.44 | 59.26 | 65.05 | 98.42 |
| monks-3 | 54.63 | 61.81 | 60.42 | 80.09 | 99.78 |
| mushroom | 100.00 | 100.00 | 100.00 | 100.00 | 99.98 |
| musk-1 | 77.31 | 84.87 | 87.39 | 89.92 | 98.86 |
| musk-2 | 97.21 | 98.61 | 98.91 | 99.27 | 99.83 |
| nursery | 99.75 | 99.91 | 99.78 | 100.00 | 99.56 |
| oocytes-merluccius-nucleus-4d | 86.27 | 83.14 | 82.35 | 83.92 | 94.29 |
| oocytes-merluccius-states-2f | 92.16 | 92.55 | 95.29 | 92.94 | 97.61 |
| oocytes-trisopterus-nucleus-2f | 81.14 | 82.02 | 79.82 | 82.46 | 96.13 |
| oocytes-trisopterus-states-5b | 93.86 | 96.05 | 93.42 | 94.74 | 99.26 |
| optical | 93.10 | 95.94 | 97.11 | 96.38 | 99.55 |
| ozone | 96.53 | 95.58 | 97.00 | 97.48 | 99.78 |
| page-blocks | 96.49 | 96.13 | 95.83 | 96.13 | 98.33 |
| parkinsons | 87.76 | 89.80 | 89.80 | 85.71 | 99.47 |
| pendigits | 94.40 | 97.11 | 97.06 | 97.37 | 99.79 |
| pima | 71.88 | 73.44 | 75.52 | 69.79 | 84.47 |
| pittsburg-bridges-MATERIAL | 88.46 | 92.31 | 88.46 | 92.31 | 99.33 |
| pittsburg-bridges-REL-L | 76.92 | 73.08 | 69.23 | 73.08 | 97.66 |
| pittsburg-bridges-SPAN | 60.87 | 69.57 | 69.57 | 73.91 | 95.09 |
| pittsburg-bridges-T-OR-D | 84.00 | 84.00 | 84.00 | 84.00 | 99.67 |
| pittsburg-bridges-TYPE | 65.38 | 65.38 | 65.38 | 57.69 | 96.39 |
| planning | 66.67 | 55.56 | 68.89 | 60.00 | 93.45 |
| plant-margin | 50.50 | 79.50 | 81.25 | 83.25 | 98.18 |
| plant-shape | 39.00 | 66.50 | 72.75 | 72.50 | 81.77 |
| plant-texture | 51.75 | 75.25 | 81.25 | 81.00 | 99.14 |
| post-operative | 40.91 | 63.64 | 72.73 | 68.18 | 95.64 |
| primary-tumor | 54.88 | 47.56 | 52.44 | 53.66 | 79.04 |
| ringnorm | 90.43 | 85.35 | 97.51 | 97.57 | 98.42 |
| seeds | 92.31 | 96.15 | 88.46 | 92.31 | 98.52 |
| semeion | 74.37 | 92.71 | 91.96 | 96.73 | 99.12 |
| soybean | 77.93 | 88.83 | 85.11 | 88.03 | 99.48 |
| spambase | 93.57 | 94.17 | 94.09 | 94.87 | 98.14 |
| spect | 62.90 | 63.44 | 63.98 | 62.37 | 90.87 |
| spectf | 91.98 | 91.98 | 49.73 | 89.30 | 99.66 |
| statlog-australian-credit | 65.12 | 63.37 | 59.88 | 61.05 | 78.57 |
| statlog-german-credit | 72.40 | 72.40 | 75.60 | 72.00 | 97.52 |
| statlog-heart | 85.07 | 91.04 | 92.54 | 92.54 | 93.88 |
| statlog-image | 95.15 | 96.88 | 95.49 | 97.75 | 99.00 |
| statlog-landsat | 87.55 | 89.25 | 91.00 | 89.90 | 96.39 |
| statlog-shuttle | 99.92 | 99.92 | 99.90 | 99.91 | 99.96 |
| statlog-vehicle | 78.67 | 77.25 | 80.09 | 81.04 | 95.04 |
| steel-plates | 73.61 | 76.49 | 78.35 | 77.53 | 94.86 |
| synthetic-control | 94.00 | 98.00 | 98.67 | 99.33 | 99.76 |

Continued on next page

| DSName | HANN15 | HANN100 | SNN | DENN | HANN100trn |
|---|---|---|---|---|---|
| teaching | 57.89 | 57.89 | 50.00 | 57.89 | 78.35 |
| thyroid | 98.37 | 98.25 | 98.16 | 98.22 | 99.65 |
| tic-tac-toe | 96.65 | 97.07 | 96.65 | 98.33 | 99.84 |
| titanic | 78.73 | 78.73 | 78.36 | 78.73 | 78.61 |
| trains | 100.00 | 50.00 | NaN | NaN | 100.00 |
| twonorm | 97.30 | 98.27 | 98.05 | 98.16 | 99.51 |
| vertebral-column-2clases | 88.31 | 85.71 | 83.12 | 85.71 | 93.85 |
| vertebral-column-3clases | 81.82 | 80.52 | 83.12 | 80.52 | 95.40 |
| wall-following | 92.45 | 94.79 | 90.98 | 91.86 | 98.16 |
| waveform | 85.84 | 84.00 | 84.80 | 83.92 | 95.14 |
| waveform-noise | 84.72 | 84.96 | 86.08 | 84.32 | 96.39 |
| wine | 97.73 | 100.00 | 97.73 | 100.00 | 99.80 |
| wine-quality-red | 62.50 | 65.00 | 63.00 | 63.50 | 90.25 |
| wine-quality-white | 54.82 | 61.03 | 63.73 | 62.25 | 81.79 |
| yeast | 59.03 | 60.65 | 63.07 | 58.22 | 68.48 |
| zoo | 96.00 | 96.00 | 92.00 | 100.00 | 99.68 |

