# OpenReview forum: "VC dimension of partially quantized neural networks in the overparametrized regime"
_ICLR.cc/2022/Conference — ICLR 2022 Poster_

### Official Review · Reviewer_QZye · 2021-11-02

**Correctness:** 4
**Technical Novelty And Significance:** 3
**Empirical Novelty And Significance:** 3
**Recommendation:** 6
**Confidence:** 2

**Main Review:**

The paper is very well written. All the concepts and ideas are explained well. The results are supported by detailed proofs and extensive references to the existing work. Nice graphics are used to present the ideas of hyperplane arrangement and resulting classifiers

The analytical results nicely complement each other. It is shown that while useful bounds on VC dimension can be proven for HANNs, the representative power of HANNs is comparable to the full precision models.

The VC dimension is known to lead to vacuous generalization bounds in full precision overparameterized neural networks. The main contribution of this paper is providing useful bounds on VC dimension for a class of highly expressive models. The bound still seems quite large though: O(k^r). It is a power of width of the layer which seems to determine the expressive capacity of the network (through determining the hyperplane arrangement). This seems a major limitation in application of the main result of the paper. If I am right, I think the paper can significantly benefit from a discussion on this (either explaining more cases where the results are useful or clarifying this limitation). For example, if we use the numbers given in experiments (which are relatively small experiments), with a data set of size 77904 and k=100, and r=3 seems to result in vacuous bounds on generalization again.

**Summary Of The Paper:**

This paper considers VC dimension for a class of partially quantized networks referred to as hyperplane augmented neural networks (HANNs). The architecture of this class of networks consists of a first hidden layer of width r and a second hidden layer of width k with Boolean (sign) activations. The name HANNs seems to be inspired by this Boolean layer which introduces a hyperplane arrangement.
An upper bound of O(k^r) is established on the VC dimension of this network that is independent of the rest of the network (a Boolean function). In addition to this result on the VC dimension of HANNs, a result is proven on their representation power. In particular, it is shown that HANNs achieve the minimax rate for classification when the posterior probability is Lipschitz.

**Summary Of The Review:**

The paper is written very well and has nice complementary results. I am however uncertain about whether the main result would be effective in explaining generalization error using VC dimension in reasonable scenarios.

---

> ### Author Response · Authors · 2021-11-17
> **VC bound**
>
> We thank the reviewer for the positive comments regarding our manuscript. We’ve made modifications to the manuscript, denoted in red.
>
> > The bound still seems quite large though: O(k^r). It is a power of width of the layer which seems to determine the expressive capacity of the network (through determining the hyperplane arrangement).
>
> The VC bound goes to zero in the minimax result. So while O(k^d), where d=r, seems large, it needs to grow only at a n^{d/(d+1)} rate, which grows more slowly than n. Thus, in the asymptotic regime as the number of samples n go to infinity and k = n^{1/(d+1)}, HANN with Theta(2^k) neurons is overparametrized with # weights >> n while the generalization error goes to zero.
>
> Regarding applying theory to practice, to the best of our knowledge no existing theoretical bound for overparametrized NNs yields "meaningful" results. Yet there is immense interest in understanding what aspects of deep NNs can explain their performance, even if the bounds aren't yet small [1,2,3,4]. Our work shows that the compressibility (in the sense of Littlestone and Warmuth [5]) of the network, as reflected by the sample compression scheme, is a useful avenue, and one that has not previously been explored -- ours is the first work applying sample compression to NNs. This seems likely to open the door to further analysis of quantized NNs.
>
> We remark that, although sharing the same name, the compression approach considered in [4] is unrelated to the traditional sample compression scheme by Littlestone and Warmuth [5] used in this work.
>
> ---
>
> [1] Bartlett, P., Foster, D., & Telgarsky, M. (2017). Spectrally-normalized margin bounds for neural networks. Advances in Neural Information Processing Systems, 30, 6241-6250.
>
> [2] Neyshabur, B., Bhojanapalli, S., Mcallester, D., & Srebro, N. (2017). Exploring Generalization in Deep Learning. Advances in Neural Information Processing Systems, 30, 5947-5956.
>
> [3] Jiang, Y., Neyshabur, B., Mobahi, H., Krishnan, D., & Bengio, S. (2019). Fantastic Generalization Measures and Where to Find Them. In International Conference on Learning Representations.
>
> [4] Arora, S., Ge, R., Neyshabur, B., & Zhang, Y. (2018). Stronger generalization bounds for deep nets via a compression approach. In International Conference on Machine Learning (pp. 254-263). PMLR.
>
> [5] Littlestone, N., & Warmuth, M. (1986). Relating data compression and learnability.

---

> > ### Comment · Reviewer_QZye · 2021-11-21
> > **Response**
> >
> > Thanks for your response. I am happy to maintain my rating of the paper.
> >
> > As a side note, I tend to disagree with "no existing theoretical bound for overparametrized NNs yields meaningful results". To me it seems there are at least two promising approaches with interesting results: neural tangent kernel and Benign overfitting. This may not be the right forum for such a discussion though.

---

### Official Review · Reviewer_UWBZ · 2021-11-02

**Correctness:** 4
**Technical Novelty And Significance:** 2
**Empirical Novelty And Significance:** 3
**Recommendation:** 6
**Confidence:** 3

**Main Review:**

To the best of my knowledge the theory appears sound and the proofs are correct and the usage of the compression scheme is neat. Moreover the experimental evaluation is thorough and surprising to some extent (e.g., how does this class achieve about state of the art performance). However, I have some reservations about the paper. First, reading the title and introduction has led me to believe that a much more general class of QNNs have small VC dimension. However the model that the authors study is basically a two layer NNs, shedding doubt on whether any non-toy quantized models will have small VC dim (could be easily checked via and experiment ala Zhang et al 2017).

Moreover, I'm doubtful about studying the VC dim of NNs in the first place. I'd love to hear the authors take of what are we aiming to learn here as in practice VC dims will give us non-meaningful bounds.

Minor comments:
Should the following be number of samples:
       Note that the above VC bound is useless in the overparameterized setting if VC(C) =(# of weights)
In proposition 4.5, P_I could be an empty set. (e.g., if a_1 = -a_2 and b_1 = -b_2). Then, there is no unique minimizer to P_I.
If m < n how can the size of J be n?

It would be helpful to further explain the second paragraph in page 8 (e.g., how did you count the weights?).

Regarding the experiments, in my opinion, using the number of weights to measure overparameterization is not indicative. A much better way to know whether the models are overparameterized is to look at the training error/loss (if it is very close to zero we are overparameterized).


**Summary Of The Paper:**

The paper studies the VC dimension and minimax properties of a subclass of partially quantized NNs they name HAC(d,r,k) where d is the input dim, r is the latent (linear map) dim and k is the boolean function dim. The authors show a bound on the VC dim of HAC of O(k^r) which is independent of the number of weights. Further, the authors show that when the conditional density p(y|x) is lipschitz, H(d, d, r) achieve minimax optimality when d is fixed and $r=n^{1/(d+2)}$.

**Summary Of The Review:**

The work is technically sound however I find the title/introduction overstating the significance of the results to some extent and not sure if the motivation is strong enough.

---

> ### Author Response · Authors · 2021-11-17
> **Relevance of VC dimension**
>
> We thank the reviewer for the constructive criticisms, and the helpful minor comments for revising our manuscript. We’ve made modifications to the manuscript, denoted in red.
>
> > First, reading the title and introduction has led me to believe that a much more general class of QNNs have small VC dimension.
>
> In the third sentence of our abstract, we modify the sentence to say "a *sub*class of partially quantized networks..." We note that HANN only requires the first layer activation to be quantized, while subsequent layers can be chosen arbitrarily. In particular, HANN includes as a subset NNs with quantizations in deeper layers, e.g., when the boolean function h is implemented as a QNN. Thus, our VC bounds also apply to them.
>
> > However the model that the authors study is basically a two layer NNs, shedding doubt on whether any non-toy quantized models will have small VC dim (could be easily checked via and experiment ala Zhang et al 2017).
>
> The models we analyzed are not required to have two layers. The boolean function h is also implemented as a NN which can have many layers. Our VC/compression bounds do not depend on the number of parameters of the NN used to implement h. In particular, the bounds hold even when h is implemented as an infinitely-wide ReLU NN of arbitrary depth.
>
> > Moreover, I'm doubtful about studying the VC dim of NNs in the first place. I'd love to hear the authors take of what are we aiming to learn here as in practice VC dims will give us non-meaningful bounds.
>
> It is widely believed that the VC dimension of NN is = number of weights. We show that, when the first layer is quantized as in HANN, the VC dimension can be much smaller. Our work suggests that quantization may be a way to improve generalization without sacrificing much in the way of expressiveness.
>
> To be frank, we believe that it is just an interesting theoretical question whether VC theory is of any use in analyzing overparametrized NNs. Our VC result answers two open theoretical questions concerning overparametrized NNs: vanishing VC bounds, and minimax rates. Bartlett et al [1] states that “it is particularly interesting to consider how the VC dimension is affected by the various attributes of the network”.
>
> VC theory is a fundamental concept in learning theory, but because existing VC dimension bounds for NNs grow at least like the number of weights, people have sought out other, more modern tools for studying generalization and minimax rates. Our work demonstrates a setting where the VC bound readily yields minimax optimality results without more sophisticated tools.
>
>
> > Should the following be number of samples: Note that the above VC bound is useless in the overparameterized setting if VC(C) =(# of weights)
>
> This sentence has been modified in our revised manuscript.
>
> > In proposition 4.5, P_I could be an empty set. (e.g., if a_1 = -a_2 and b_1 = -b_2). Then, there is no unique minimizer to P_I. If m < n how can the size of J be n?
>
> We’ve clarified by adding the additional requirement that P_{[m]} be nonempty, which is met in the subsequent application of the proposition. We also modified the statement to say that |J| = min{m, n}. The proposition only says something interesting when m > n. When m ≤ n, the proposition is trivially satisfied by setting J = [m]. Both the proposition and its proof have been updated.
>
> > It would be helpful to further explain the second paragraph in page 8 (e.g., how did you count the weights?).
>
> We added the explanation to Appendix C with a pointer to it in the main article’s experimental section.
>
> > Regarding the experiments, in my opinion, using the number of weights to measure overparameterization is not indicative. A much better way to know whether the models are overparameterized is to look at the training error/loss (if it is very close to zero we are overparameterized).
>
> We use early stopping where the training error does not always go to zero. Even with early stopping, HANN100 reaches ≥ 99% training accuracy on 41 out of 121 datasets. Meanwhile over the same 41 datasets, the test accuracy is 89.8%, when averaged over the three tested models (HANN, SNN, and DENN). We added an additional column for the training accuracy to the Table in Appendix E with this additional information.
>
> ---
>
> [1] Bartlett, P. L., Harvey, N., Liaw, C., & Mehrabian, A. (2019). Nearly-tight VC-dimension and pseudodimension bounds for piecewise linear neural networks. The Journal of Machine Learning Research, 20(1), 2285-2301.

---

> > ### Comment · Reviewer_UWBZ · 2021-11-24
> > **Response to rebuttal**
> >
> > I would like to thanks the authors for addressing most of my concerns and raised my score accordingly. I would not mind seeing the paper published in ICLR however I'm not willing to give it a higher score since I'm still unconvinced whether VC dim is the right point of focus when trying to understand generalization of DNNs.

---

### Official Review · Reviewer_W4WM · 2021-11-03

**Correctness:** 3
**Technical Novelty And Significance:** 3
**Empirical Novelty And Significance:** 4
**Recommendation:** 8
**Confidence:** 2

**Main Review:**

Strength:
- demystify the generalization puzzle of overparameterized neural networks is an important topic to study
- novelty: the first VC theory of overparameterized neural networks
- the construction is not too complicated

Weakness:
- I don't fully understand the implication of Theorem 4.2. size(rho, kappa) critically depends on r, the rank of the weight matrix W. But if we decompose W=UV, where U is d*r and V=r*k. Isn't the VC dimension of HANN still higher than the number of effective parameters, dr+rk? In this case, what benefit do HANN have over ReLU and sigmoid networks?

Post rebuttal
====

Thanks for answering my questions. I'm still confused about what role does the binary activation play here though. I think the critical assumption of Theorem 4.2 is that the weight matrix is low-rank. In this case, can't I just reformulate the layer with low-rank weight as two layers with a small bottleneck hidden layer of dimensionality r, which has much smaller number of parameters than the equivalent network? If this is the case, I think the result is only relevant to the low-rank assumption, not the HANN itself.

I'm still maintaining my score based on these concerns.

**Summary Of The Paper:**

This paper proposes a network architecture, HANN. HANN contains a hidden layer of binary activations, and can has a VC dimension smaller than the number of network parameters. HANN can be potentially a VC theory for studying the generalization of overparameterized neural networks. HANN is competitive with sota on UCI datasets.

**Summary Of The Review:**

This paper solves an important problem and is novel, though I am not fully convinced by the claims due to some potential misunderstanding.

---

> ### Author Response · Authors · 2021-11-17
> **Effective parameters of HANN**
>
> We thank the reviewer for pointing out the strengths and for drawing our attention to areas that require clarification. We’ve made modifications to the manuscript, denoted in red.
>
> > I don't fully understand the implication of Theorem 4.2. size(rho, kappa) critically depends on r, the rank of the weight matrix W. But if we decompose W=UV, where U is dr and V=rk. Isn't the VC dimension of HANN still higher than the number of effective parameters, dr+rk?
>
> In Theorem 4.2, both the weight matrix W and the boolean function h are to be compressed/reconstructed. Thus, both W and h contribute to the number of effective parameters. We note that size(rho, kappa) does not depend on the number of parameters used to implement the boolean function h, which can be as high as one wishes. For instance, the same compression/VC bound holds even when h is implemented using an infinitely-wide ReLU NN. Thus, the VC dimension of HANN is much lower than the number of effective parameters.
>
> > In this case, what benefit do HANN have over ReLU and sigmoid networks?
>
> The main benefit, from a VC theory perspective, is that the 1-bit sign-activation in the bottom-layer dramatically reduces the VC dimension. On the other hand, using ReLU or sigmoid activation, the VC dimension is much larger, on the order of the number of parameters of the network.

---

> ### Author Response · Authors · 2021-11-26
> **Response to Post rebuttal**
>
> We thank the reviewer for the post rebuttal follow-up and are happy to further clarify.
>
> The low rank $W$ can indeed be converted to two layers with the number of parameters as the reviewer mentioned. However, this accounts only for the parameters in the "$W$" portion of the overall network, namely the $sign(W'x +b)$ portion. The other portion of the network, the boolean function $h$, is also parametrized by and trained as a neural network (recall that $h$ is applied to $sign(W'x +b)$). Thus the parameters used in $h$ add to the parameter count of the overall HANN as well. The key idea is that, regardless of how many parameters $h$ has, only $\binom{k}{\le r}$ samples are required for the sample compression scheme in the proof of Theorem 4.2 to reconstruct $h$. This is made possible with the $sign$-activation and the result in hyperplane arrangements due to Buck (1943) (see Eq (1) on our page 3).

---

> > ### Comment · Reviewer_W4WM · 2021-11-27
> > **Thanks for the clarifications**
> >
> > Thanks for the clarifications. These addressed my concerns. I am raising my score to 7.

---

### Decision · Program_Chairs · 2022-01-20

**Decision:**

Accept (Poster)

**Comment:**

The authors analyzing the VC-dimension of a class of neural networks
with hard thresholds at the hidden nodes that include a low-rank weight
matrix and hard-thresholds at hidden units.  The bounds are independent
of the number of weights used to represent functions mapping a hidden
layer to the output.  They also provided some experiments supporting the
practicality of networks like those treated in their theoretical analysis.

There was some question about whether the VC-dimension continues to be relevant.
Also, while the upper bounds have attractive properties that were highlighted by the
authors, they also are not very strong in other respects.

The consensus view overall, though, was that this is a "nice result",
a clean illustration of a generalization affect of the type that has
been of wide interest lately.